# Psychological Distress and Food Insecurity among International Students at a Hungarian University: A Post-Pandemic Survey

**DOI:** 10.3390/nu16020241

**Published:** 2024-01-12

**Authors:** Soukaina Hilal, László Róbert Kolozsvári, Putu Ayu Indrayathi, Sami Najmaddin Saeed, Imre Rurik

**Affiliations:** 1Department of Family and Occupational Medicine, University of Debrecen, 4032 Debrecen, Hungary; kolozsvari.laszlo@med.unideb.hu (L.R.K.); indrayathi.ayu@med.unideb.hu (P.A.I.); 2Doctoral School of Health Sciences, University of Debrecen, 4032 Debrecen, Hungary; saeed.sami@med.unideb.hu (S.N.S.); rurik.imre@med.unideb.hu (I.R.); 3Department of Public Health and Epidemiology, University of Debrecen, 4028 Debrecen, Hungary; 4Department of Family Medicine, Semmelweis University, 1085 Budapest, Hungary

**Keywords:** anxiety, food insecurity, stress, depression, international students, post-pandemic, Hungary

## Abstract

The aim of the present study was two-fold: Firstly, to estimate the prevalence of psychological distress among international students at a Hungarian university two years after the COVID-19 outbreak; and secondly, to identify its demographic and socioeconomic factors, with special regard to the students’ food-security status. A cross-sectional study using a self-administered questionnaire was carried out from 27 March to 3 July 2022 among international students at the University of Debrecen. The questionnaire included information on demographic and socioeconomic characteristics, food-security status (six-item United States Department of Agriculture Food Security Survey Module (USDA-FSSM)), and psychological distress (Depression, Anxiety and Stress Scale (DASS-21)). Bivariate analysis was conducted to examine the potential associations between demographic/socioeconomic factors and psychological distress. Additionally, multiple logistic regression was employed to further analyze these associations. Of 398 participants, 42.2%, 48.7%, and 29.4% reported mild to extremely severe depression, anxiety, and stress, respectively. The ages 18–24 (AOR = 2.619; 95% CI: 1.206–5.689) and 25–29 (AOR = 2.663; 95% CI: 1.159–6.119), reporting a low perception of health status (AOR = 1.726; 95% CI: 1.081–2.755), and being food insecure (AOR = 1.984; 95% CI: 1.274–3.090) were significantly associated with depressive symptoms. Being female (AOR = 1.674; 95% CI: 1.090–2.571), reporting a low perception of health status (AOR = 1.736; 95% CI: 1.098–2.744), and being food insecure (AOR = 2.047; 95% CI: 1.327–3.157) were significantly associated with anxiety symptoms. Furthermore, being female (AOR = 1.702; 95% CI: 1.026–2.824)), living with roommates (AOR = 1.977; 95% CI: 1.075–3.635), reporting a low perception of health status (AOR = 2.840; 95% CI: 1.678–4.807), and being food insecure (AOR = 2.295; 95% CI:1.398–3.767) were significantly associated with symptoms of stress. Psychosocial programs combined with strategies to alleviate food insecurity are required to enhance international students’ mental health and well-being.

## 1. Introduction

Over the last few years, there has been a substantial growth in the number of international students worldwide. For instance, the population of international students in Hungarian universities and colleges has nearly doubled between 2008 and 2017 [1]. Recent statistics from the Hungarian Education Authority show that 37,925 international students were enrolled in Hungarian universities during the 2020/2021 academic year [2]. These students contribute significantly to Hungary’s economy as well as its social and cultural diversity [3].

While studying abroad has the potential to benefit international students’ personal and academic development, it also comes with a host of challenges, including language difficulties, processes of acculturation, a lack of social support, potential discrimination or racism, and financial insecurity. These challenges, combined with the normative stressors of starting and attending a higher educational institution, can take a toll on international students’ mental health [4]. Unfortunately, international students may encounter several barriers (e.g., language and cultural differences) that hinder them from seeking professional help, resulting in further mental health challenges [5]. Furthermore, the COVID-19 pandemic might have amplified these challenges among international students, increasing their vulnerability to inequalities and mental health issues [6]. Indeed, a recent longitudinal study comparing the psychological health of international students before versus after the pandemic has reported sizeable increases in the rates of self-reported anxiety and depression [7].

Previous research indicates that international students may be more susceptible to facing financial hardship [8]. This vulnerability was further escalated by the loss of jobs and income during the COVID-19 pandemic, which resulted in many international students struggling to afford basic necessities, including rent and food supplies [6]. These financial challenges may, therefore, have heightened the risk of food insecurity for international students. Indeed, recent post-pandemic studies reported greater levels of food insecurity among international students [9,10]. However, only a small number of studies have addressed the issue of food insecurity among international students in European countries, and none have been carried out in Hungary. Hungarian higher education has attracted international students by offering various scholarships, such as the Erasmus and Stipendium Hungaricum scholarship programs. While these programs provide valuable opportunities for students seeking academic and cultural experiences, not all of them cover the full range of living expenses [11]. In addition, international students studying in Hungary have a limited number of working hours due to their study permit conditions [12]. Altogether, these factors may increase their vulnerability to financial challenges and food insecurity.

Food insecurity is a major issue that has been linked to a myriad of physical and mental health conditions among university students [13]. Numerous studies have reported that students facing food insecurity had higher odds of reporting unhealthy diets, altered sleep quality, stress, anxiety, depression, and poor academic performance relative to their food-secure counterparts [14,15,16,17]. However, the available literature on these associations among international students is limited [18], despite the challenges they face that could elevate their susceptibility to food insecurity, thereby impacting their health and mental well-being [19,20].

Most studies evaluating the emotional states of international students amid the COVID-19 pandemic were conducted during its early stages. Likewise, within the Hungarian context, research on the psychological health of international students is scarce, with all existing studies conducted thus far focusing solely on the pandemic’s initial phase. Given the pandemic’s long duration and the persistence of the stressors impacting international students’ mental health, it remains essential to consistently monitor their mental health.

Our study was conducted to estimate the prevalence of three forms of psychological distress in the further phase of the COVID-19 pandemic and to explore their demographic and socioeconomic factors with special regard to their food-security status.

## 2. Materials and Methods

### 2.1. Study Setting, Design, and Participants

A cross-sectional study utilizing a self-administered questionnaire was carried out among international students who were enrolled at the University of Debrecen (UniDeb), Hungary, during the second semester of the 2021/2022 academic year. UniDeb, which is situated in the city of Debrecen, is one of the largest higher educational institutions in Hungary. The university has 13 faculties offering more than 100 programs at the undergraduate, postgraduate, and research levels [21]. Undergraduate programs’ length of study ranges from four years (i.e., Engineering, Agriculture, and Business) to six years (i.e., General Medicine), while postgraduate programs usually last two years for Master’s degrees and four years for PhD/Doctoral degrees.

Using a non-probability “convenience” sampling technique, self-reported data were collected from 27 March to 3 July 2022. Eligible participants were international students (aged ≥ 18 years) enrolled in a full-time undergraduate or postgraduate degree program at any of the faculties of UniDeb. Students participating in Erasmus or other mobility programs were excluded. The researchers invited and approached the students in the foyer of each faculty to voluntarily participate in the study. After screening for eligibility criteria through a brief verbal interview, eligible participants were given a hard copy of a booklet that included an information letter outlining the objectives of our study, a consent form, and the questionnaire. Once completed, students placed the completed questionnaire booklet into the collecting boxes. The students were assured that anonymity and confidentiality would be maintained, and that they had the freedom to exit the study whenever they chose. All the questionnaires were administered in English, the language of instruction for international students at UniDeb.

### 2.2. Sample Size Calculation

The required sample size for this study was computed utilizing the Raosoft online calculator based on a 95% confidence level, 5% error rate, and a response distribution of 50% [22]. According to the university’s official website, around 7000 international students were registered for the academic year 2021/2022 [21]. Thus, the minimum sample size was determined to be 365. Taking into consideration an anticipated 10% rate of incomplete or non-responses, the final sample size was estimated to be 401 participants.

### 2.3. Study Measures

A self-administered questionnaire was constructed using previously tested instruments. The outcome variable was psychological distress, and it was evaluated utilizing the validated English version of the Depression, Anxiety, and Stress Scale (DASS-21) [23]. This 21-item survey is composed of three independent subscales: DASS-D for depression (seven questions), DASS-A for anxiety (seven questions), and DASS-S for stress (seven questions). The DASS-21 utilizes a four-point Likert scale (0–3) to evaluate participants’ emotional symptoms over the preceding week. For each subscale, the overall score is determined by summing the scores of all seven questions and then multiplying them by 2. As a severity measure, the DASS-21 is used to determine different severity levels. For DASS-D, cut-off points of 9 or lower reflect normal, 10–13 mild, 14–20 moderate, 21–27 severe, and 28 or higher for extremely severe depressive symptoms, respectively. For DASS-A, cut-off points of 7 or lower reflect normal, 8–9 mild, 10–14 moderate, 15–19 severe, and 21 or higher for extremely severe anxiety symptoms, respectively. For DASS-S, cut-off points of 14 or lower reflect normal, 15–18 mild, 19–25 moderate, 26–33 severe, and 35 or higher for extremely severe stress symptoms, respectively [23]. The Cronbach’s alpha for each subscale and the whole scale were as follows: 0.839, 0.814, 0.900, and 0.936, indicating high reliability.

The independent variable of primary interest was food insecurity, which was assessed using the US Department of Agriculture Food Security Survey Module (USDA-FSSM). This self-report survey consists of six items addressing people’s difficulties in accessing adequate food because of financial constraints. The highest potential score on this survey is 6 and the lowest potential score is 0. Scores are obtained by summing the number of affirmative answers. Respondents who scored 2 or higher were classified as having food insecurity based on the USDA guidelines [24]. The Cronbach’s alpha was 0.865.

The following independent variables were additionally incorporated into our analysis: age, gender (male or female), region of origin (Asia, Africa, Europe, or the Americas), religion (yes or no), relationship status (single or partnered), living arrangement (lived alone, with roommates, or with family), level of study (undergraduate or postgraduate), field of study (healthcare or other), sponsorship (scholarship or self-sponsor), employment (yes or no), length of stay in Hungary (<1 year, 2–4, or ≥5 years), Hungarian language for communication with native people (good, moderate, or poor), tobacco use (yes or no), alcohol consumption (never or occasional, once a week, more than once a week), self-rated health status (poor or good). Participants were given the following options to evaluate their communication in the host language: very poor, poor, moderate, good, and very good. The responses were collapsed into “poor”, “medium”, and “good”. Regarding self-rated health, participants were asked: “How do you evaluate your general health status?” The response options were: poor, average, good, very good, or excellent. Given the distribution of responses, the responses “poor” and “average” were categorized as “poor”, while all remaining responses were categorized as “Good”.

### 2.4. Analysis

After entering the collected data into an Excel spreadsheet, they were analyzed using IBM SPSS 26 (IBM Corp., Armonk, NY, USA). Qualitative variables were expressed using frequencies (*n*) and percentages (%), whereas continuous variables were represented as the median along with the interquartile range (IQR) for data that do not follow a Gaussian distribution. The normality criterion was analyzed using the Shapiro–Wilk and Kolmogorov–Smirnov tests. First, bivariate analyses were conducted using the Fisher exact or chi-square tests. Then, the variables that showed a value of *p* < 0.20 in bivariate analyses were selected as candidate variables for multiple regression analyses (Appendix A). Finally, multiple logistic regression analyses were run using the Enter method. Our results were outlined utilizing adjusted odds ratios (AORs) along with 95% confidence intervals (CIs), at a *p*-value less than 0.05. The models’ fit was evaluated utilizing the Hosmer–Lemeshow test.

### 2.5. Ethical Considerations

Our study complied with the ethical standards of the Declaration of Helsinki and secured approval from the Regional Institutional Research Ethics Committee, Clinical Centre, University of Debrecen (approval number: DE RKEB/IKEB 5933-2021). All participants were required to provide consent before filling out the questionnaire.

## 3. Results

### 3.1. Background Characteristics

Out of 442 international students who agreed to participate, 419 responded to our questionnaire, yielding a response rate of 94.8%. However, 21 questionnaires were excluded due to incompleteness. Therefore, the remaining 398 were included in our analysis.

Participants’ ages ranged from 18 to 50 years (median: 23, interquartile range (IQR) 22–26), with 62.6% of them aged 18 to 24 years (Table 1). There was a slight predominance of females (53.8%) compared to males (46.2%). The majority were from Asia (63.3%), followed by Africa (25.6%), Europe (6.0%), and the Americas (6.0%). Most participants identified as religious (78.6%) and did not have a partner (70.6%). Around 40% of the participants lived with roommates during their studies. The majority of students were undergraduates (73.4%) and enrolled in a healthcare program (63.8%). In addition, 41.7% reported receiving a scholarship, and 11.8% had a side job. Moreover, 45% of the students had stayed in Hungary for more than four years. Regarding language proficiency, 76% of the participants rated their communication in Hungarian as poor, 18.6% as average, and 5.3% as good. Finally, 30.7% reported a poor health status, and 40.2% reported food insecurity.

### 3.2. Prevalence of Psychological Distress

Table 2 lists the prevalence of psychological distress among the participants.

Among the types of psychological distress examined, anxiety was the most common among the participants, with a prevalence of 48.6%, followed by depression (42.2%) and stress (29.4%). Moderate depressive, anxiety, and stress symptoms were reported in 14.8%, 20.1%, and 10.8% of respondents, respectively (Table 2).

### 3.3. Factors Associated with Psychological Distress among the Participants According to the Bivariate Analysis

Appendix A shows the bivariate analysis of factors associated with psychological distress. Depressive symptoms were most common in international students with the following characteristics: being in the 25–29 years age group (ꭓ2 = 6.481, *p* = 0.039), not having a partner (ꭓ2 = 5.354, *p* = 0.021), having poor health (ꭓ2 = 5.345, *p* = 0.021), and being food insecure (ꭓ2 = 14.605, *p* < 0.001). Anxiety symptoms were most common in international students with the following characteristics: being female (ꭓ2 = 7.079, *p* = 0.008), living with roommates (ꭓ2 = 7.136, *p* = 0.028), having poor health (ꭓ2 = 7.801, *p* = 0.005), and being food insecure (ꭓ2 = 17.435, *p* < 0.001). Stress symptoms were most common in international students with the following characteristics: being female (ꭓ2 = 8.345, *p* = 0.004), not having a partner (ꭓ2 = 5.148, *p* = 0.023), sharing accommodation with roommates (ꭓ2 = 12.419, *p* = 0.002), having spent <1 year in Hungary (ꭓ2 = 6.257, *p* = 0.044), having poor Hungarian proficiency (ꭓ2 = 10.195, *p* = 0.006), having poor health (ꭓ2 = 16.722, *p* < 0.001), and being food insecure (ꭓ2 = 17.435, *p* < 0.001).

### 3.4. Factors Associated with Psychological Distress among the Participants According to the Multivariable Logistic Regression Analyses

Variables that showed a value of *p* < 0.20 in bivariate analyses were considered for multivariable analyses: 9 variables for depression (age, relationship status, living arrangements, field of study, length of stay in Hungary, communication in Hungarian, tobacco use, self-perceived health, and food-security status), 7 variables for anxiety (gender, religion, living arrangements, communication in Hungarian, tobacco use, self-perceived health, and food-security status), and 11 variables for stress (age, gender, region of origin, relationship status, living arrangements, education level, field of study, length of stay in Hungary, communication in Hungarian, self-perceived health, and food-security status).

As shown in Table 3, regression analysis indicated that an age of 18–24 years (AOR = 2.62, 95% CI = 1.21–5.69, *p* = 0.015), and age 25–29 years (AOR = 2.66, 95% CI = 1.16–6.12, *p* = 0.021) were significantly associated with experiencing symptoms of depression. Reporting poor health was associated with greater odds of symptoms of depression (AOR = 1.726, 95% CI = 1.081–2.755, *p* = 0.022). Moreover, students with food insecurity reported higher odds of depressive symptoms than food secure students (AOR = 1.984, 95% CI = 1.274–3.090, *p* = 0.002).

In terms of anxiety, our results showed that female participants were 1.67 times more likely to report anxiety relative to their male counterparts (AOR = 1.67, 95% CI = 1.09–2.57, *p* = 0.019). Reporting a poor self-rated health status was associated with greater odds of anxiety (AOR = 1.736, 95% CI = 1.098–2.744, *p* = 0.018). Likewise, those who reported food insecurity had higher odds of anxiety symptoms (AOR = 2.047, 95% CI = 1.327–3.157, *p* = 0.001) (Table 4).

Factors associated with symptoms of stress include being a female (AOR = 1.702, 95% CI = 1.026–2.824, *p* = 0.039), coming from a European country (AOR = 2.982, 95% CI = 1.068–8.320, *p* = 0.037), sharing accommodation with roommates (AOR = 1.977, 95% CI = 1.075–3.635, *p* = 0.028), having a poor self-rated health status (AOR = 2.840, 95% CI = 1.678–4.807, *p* < 0.001), and experiencing food insecurity (AOR = 2.295, 95% CI = 1.398–3.767, *p* = 0.001) (Table 5).

## 4. Discussion

The current study found a considerable degree of psychological distress among international students at the University of Debrecen, Hungary. Overall, 42.2%, 48.7%, and 29.4% of the students reported depression, anxiety, and stress symptoms, respectively. These rates are similar to those observed in previous studies during the initial phases of the COVID-19 pandemic. For example, a previous study conducted in July 2020 among international students studying in China utilizing the same assessment tool (DASS-21) and cut-offs found that 43.6%, 24.6%, and 38.8% had depression, anxiety, and stress symptoms [25]. Another study performed in December 2020 among international students living in South Korea indicated that 49% and 39.6% reported depression and anxiety, respectively [26]. Moreover, the rates of self-reported depression and anxiety among international students in the current study appear to be higher than those observed among the Hungarian population during the first year of the pandemic, which found that 34.1% were depressed and 36.2% were anxious [27]. It is concerning that in the second year of the pandemic after most COVID-19 restrictions had been lifted and the vaccines were available for general use, a significant proportion of the students reported elevated rates of psychological distress. Care strategies must be designed in response to these alarming rates. The University of Debrecen provides free psychological counselling services for international students. These services are offered by qualified psychologists. Ensuring that international students are informed about the availability of these support systems can serve as a foundation for addressing their mental health needs.

In this study, 40.2% of our participants reported being food insecure. This finding is consistent with earlier studies carried out during the pandemic. For instance, a study by Ibiyemi et al. in the US yielded a comparable prevalence of food insecurity (32%) among international students [28]. However, another cross-sectional online survey recorded a lower prevalence of food insecurity (18.7%) among international students in Australia [9]. These variations can be attributed to the use of different instruments, sampling methods, and sample sizes. It is also worth noting that the rate of food insecurity in our study is remarkably higher than the national rate of food insecurity in Hungary [29]. This finding indicates the necessity for tailored interventions to alleviate food insecurity among this population.

Our study demonstrated that participants with younger ages were more likely to be depressed. This was consistent with previous studies, such as one conducted in India among dental students and practitioners, which found that those who were younger than 30 years old were more likely to be depressed [30]. This may be because younger students, as they are going from adolescence to early adulthood, are in a particularly vulnerable phase for the onset of depression, added to the process of adaptation in a foreign country [31].

In our study, female participants exhibited higher levels of stress and anxiety symptoms relative to male participants. This finding coincides with several earlier studies [32,33], which also observed higher rates of stress and anxiety among women. This may be attributed to the biological, hormonal, and psychosocial differences between males and females [34].

Consistent with some previous findings [35,36], this study found that students who shared accommodation with roommates were more likely to report stress symptoms. While studying in Hungary, international students often share accommodation with roommates from different cultural and language backgrounds. These differences can sometimes lead to communication misunderstandings or conflicts, which can negatively affect their psychological health [37]. In addition, the uncertainties about health and safety measures caused by the COVID-19 pandemic may have amplified these challenges, resulting in further stress symptoms.

In addition, the odds of the three forms of psychological distress examined in our study were significantly increased for participants who reported a poor health condition. A previous study investigating the psychological effects of the pandemic on a sample of Chinese international students studying in the USA found that those who rated their health status as poor experienced higher rates of stress, anxiety, and depression [38]. Thus, it is imperative that healthcare professionals offer resources for psychological support for international students with a poor health status.

Finally, students experiencing food insecurity reported greater odds of depression, anxiety, and stress than their food-secure peers. This finding was in accordance with several recent post-pandemic studies [10,39]. For instance, a cross-sectional survey of local and international students attending one of the Australian universities in 2020 found that students who reported being food insecure had 8.07-times-increased odds of psychological distress compared with those who were food secure [10]. Our results were also consistent with a large body of prior research showing an association between food insecurity and negative emotional states [40,41,42]. Although the present study cannot ascertain a causal link between food insecurity and psychological distress, it has been suggested that the inability to access and maintain enough food supplies generates a stress response that could lead to the onset or exacerbation of mental health issues. In addition, food insecurity has also been found to be associated with unhealthy dietary patterns, which, in turn, can have adverse effects on mental health [43] 

As per our knowledge, this was the first study to estimate the prevalence of psychological distress symptoms among international students in Hungary after the COVID-19 pandemic. In addition, we determined a number of demographic and socioeconomic factors associated with these symptoms. However, it is crucial to acknowledge certain limitations when interpreting our findings. First, our study used a convenience sampling method, which could potentially introduce selection bias. Second, the DASS-21 questionnaire is suitable for screening psychological distress symptoms but cannot be used as a diagnostic measure. Third, the cross-sectional nature of the study cannot confirm causality. Fourth, this study was performed in a single university; therefore, the finding cannot be generalized to other institutions in the country. Fifth, this study was carried out during the exam period, which could have increased the reporting of the symptoms of psychological distress. Lastly, participants from Asian countries were overrepresented, which might have skewed the results.

## 5. Conclusions

The present study reveals alarming levels of psychological distress among international students at the University of Debrecen. Factors including age, gender, living arrangements, self-rated health, and food insecurity status were strongly associated with psychological distress in our population. These findings should draw the attention of university authorities to design appropriate interventions and to provide adequate well-being support for this vulnerable population.

## Figures and Tables

**Table 1 nutrients-16-00241-t001:** Participants’ characteristics (*n* = 398).

Variables		*n* (%)
Age	Median (IQR)	23.0 (4.0)
Age (years)	18–24	249 (62.6)
	25–29	102 (25.6)
	≥30	27 (11.8)
Gender	Male	184 (46.2)
	Female	214 (53.8)
Region of origin	Asia	252 (63.3)
	Africa	102 (25.6)
	Europe	24 (6.0)
	The Americas	20 (5.0)
Religious	No	85 (21.4)
	Yes	313 (78.6)
Relationship status	Single	281 (70.6)
	Partnered	117 (29.4)
Living arrangement	Living alone	154 (38.7)
	Living with roommate(s)	160 (40.2)
	Living with family	84 (21.1)
Education level	Undergraduate	292 (73.4)
	Postgraduate	106 (26.6)
Field of study	Healthcare	254 (63.8)
	Other	144 (36.2)
Sponsorship	Scholarship	166 (41.7)
	Self-sponsor	232 (58.3)
Employment	No	351 (88.2)
	Yes	47 (11.8)
Length of stay (years)	<1	98 (24.6)
	1–4	119 (29.9)
	≥5	181 (45.5)
Communication in Hungarian	Poor	303 (76.1)
	Medium	74 (18.6)
	Good	21 (5.3)
Tobacco	No	342 (85.9)
	Yes	56 (14.1)
Alcohol consumption	Never/occasional	337 (84.7)
	Once a week	50 (12.6)
	More than once a week	11 (2.7)
Self-rated health status	Good	276 (69.3)
	Poor	122 (30.7)
Food-security status	Food-secure	238 (59.8)
	Food-insecure	160 (40.2)

**Table 2 nutrients-16-00241-t002:** Severity of psychological distress based on the DASS-21 scale’s categorization.

	Depression *n* (%)	Anxiety *n* (%)	Stress *n* (%)
Normal	230 (57.8)	205 (51.5)	281 (70.6)
Mild	49 (12.3)	42 (10.5)	42 (10.6)
Moderate	59 (14.8)	80 (20.1)	43 (10.8)
Severe	31 (7.8)	21 (5.3)	22 (5.5)
Extremely severe	29 (7.3)	50 (12.6)	10 (2.5)

**Table 3 nutrients-16-00241-t003:** Factors associated with symptoms of depression.

Variables	Presence of Depression *n* (%)	AOR (95% CI)	*p*
Age			
18–24 years	10 (64.3)	2.619 (1.206–5.689)	0.015 *
25–29 years	48 (28.6)	2.663 (1.159–6.119)	0.021 *
≥30 years	12 (7.1)	Ref	
Relationship status			
Single	129 (45.9)	Ref	
Partnered	39 (33.3)	0.702 (0.428–1.152)	0.161
Living arrangements			
Alone	56 (36.4)	Ref	
Roommate(s)	78 (51.2)	1.403 (0.830–2.372)	0.207
Family	34 (40.5)	1.335 (0.737–2.416)	0.340
Field of study			
Healthcare	100 (39.4)	Ref	
Other	68 (47.2)	1.383 (0.715–2.674)	0.335
Length of stay in Hungary			
<1 year	51 (52.0)	Ref	
1–4 years	39 (32.8)	0.6559 (0.239–1.066)	0.078
≥5 years	78 (43.1)	1.024 (0.476–2.203)	0.951
Communication in Hungarian			
Poor	137 (45.2)	Ref	
Medium	25 (33.8)	0.602 (0.338–1.074)	0.086
Good	6 (28.6)	0.572 (0.202–1.619)	0.293
Tobacco use			
No	139 (40.6)	Ref	
Yes	29 (51.8)	1.460 (0.787–2.709)	0.230
Self-rated health status			
Good	106 (38.4)	Ref	
Poor	62 (50.8)	1.726 (1.081–2.755)	0.022 *
Food-security status			
Food secure	82 (34.5)	Ref	
Food insecure	86 (53.8)	1.984 (1.274–3.090)	0.002 *

Ref = a reference category; * Significant at *p* < 0.05.

**Table 4 nutrients-16-00241-t004:** Factors associated with symptoms of anxiety.

Variables	Presence of Anxiety *n* (%)	AOR (95% CI)	*p*
Gender			
Male	76 (41.3)	Ref	
Female	117 (54.7)	1.674 (1.090–2.571)	0.019 *
Religious			
Yes	158 (50.5)	Ref	
No	35 (41.2)	0.648 (0.386–1.090)	0.102
Living arrangements			
Alone	62 (40.3)	Ref	
Roommate(s)	88 (55.0)	1.498 (0.931–2.412)	0.096
Family	43 (51.2)	1.518 (0.864–2.666)	0.146
Communication in Hungarian			
Poor	152 (50.2)	Ref	
Medium	35 (47.3)	0.928 (0.541–1.593)	0.787
Good	6 (28.6)	0.584 (0.207–1.649)	0.310
Tobacco use			
No	161 (47.1)	Ref	
Yes	32 (57.1)	1.687 (0.913–3.116)	0.095
Self-rated health status			
Good	121 (43.8)	Ref	
Poor	72 (59.0)	1.736 (1.098–2.744)	0.018 *
Food-security status			
Food secure	95(39.9)	Ref	
Food insecure	98 (61.3)	2.047 (1.327–3.157)	0.001 *

Ref = a reference category; * Significant at *p* < 0.05.

**Table 5 nutrients-16-00241-t005:** Factors associated with symptoms of stress.

Variables	Presence of Stress *n* (%)	AOR (95% CI)	*p*
Age			
18–24 years	72 (28.9)	2.228 (0.847–5.862)	0.105
25–29 years	36 (35.3)	1.970 (0.750–5.177)	0.169
≥30 years	9 (19.1)	Ref	
Gender			
Male	41 (22.3)	Ref	
Female	76 (35.5)	1.702 (1.026–2.824)	0.039 *
Region of origin			
Asia	67 (26.6)	Ref	
Africa	31 (30.4)	0.998 (0.560–1.777)	0.994
Europe	10 (41.7)	2.982 (1.068–8.320)	0.037 *
Americas	9 (45.0)	2.443 (0.836–7.137)	0.103
Relationship status			
Single	92 (32.7)	Ref	
Partnered	25 (21.4)	0.579 (0.322–1.040)	0.068
Living arrangements			
Alone	32 (20.8)	Ref	
Roommate(s)	62 (38.8)	1.977 (1.075–3.635)	0.028 *
Family	23 (27.4)	1.425 (0.711–2.855)	0.318
Education level			
Undergraduate	79 (27.1)	Ref	
Postgraduate	38 (35.8)	1.172 (0.531–2.588)	0.694
Field of study			
Healthcare	67 (26.4)	Ref	
Other	50 (34.7)	1.022 (0.461–2.264)	0.958
Length of stay in Hungary			
<1 year	38 (38.8)	Ref	
1–4 years	28 (23.5)	0.598 (0.295–1.216)	0.156
≥5 years	51 (28.2)	0.899 (0.379–2.134)	0.810
Communication in Hungarian			
Poor	100 (33.0)	Ref	
Medium	16 (21.6)	0.507 (0.254–1.013)	0.054
Good	1 (4.8)	0.134 (0.016–1.104)	0.062
Self-rated health status			
Good	64 (23.2)	Ref	
Poor	53 (43.4)	2.840 (1.678–4.807)	<0.001 *
Food-security status			
Food secure	50 (21.0)	Ref	
Food insecure	67 (41.9)	2.295 (1.398–3.767)	0.001 *

Ref = a reference category; * Significant at *p* < 0.05.

## Data Availability

The data presented in this study are available on request from the corresponding author. The data are not publicly available due to IRB restrictions.

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
