# Peer review of "Psychological Distress and Food Insecurity among International Students at a Hungarian University: A Post-Pandemic Survey"

_nutrients, 2024, doi:10.3390/nu16020241_

Round 1

Reviewer 1 Report

Comments and Suggestions for Authors

Overall, this was an interesting paper on an important topic.  I have feedback on how to improve it:

- Why did you focus on Hungary?  (I'm not saying you should have focused on a different country, merely that you should justify your choice.)

- What are the national rates of depression, anxiety, and food insecurity in Hungary?  This would be a good baseline for you to compare against the rates among international students.

- Is the US Department of Agriculture Food Security Survey Module validated for use in Hungary?

- Why do you think that students from elsewhere in Europe reported higher levels of stress than students from farther away?

- Why did you ask participants if they were religious?  There's research that looks at how religiosity can mitigate depression and anxiety outcomes, so I'm curious to know if your participants experienced that as well, since so many of your participants reported being religious.

- The two studies you cite in the Discussion (in China in July 2020 and in South Korea in December 2020) were both conducted early in the COVID-19 pandemic.  There's plenty of research on how COVID impacts mental health; did the articles you cited mention that as a factor impacting mental health outcomes?  I can imagine that COVID would be especially impactful on students who are studying away from home.

- Did you run any tests on food insecurity and employment status?  If the international students have a limited income due to being on a visa, then they might not be able to afford much food. 

- A future study could incorporate questions on discrimination, since that can impact mental health.  (E.g., did international students face any discrimination for being from another part of the world?)

- A future study could also look at how close international students lived to affordable food stores and/or restaurants.  If they live far away from stores and restaurants and they don't have a reliable source of transportation, their food options might be limited.

Author Response

Dear reviewer,
Thank you for taking the time to review our manuscript “Psychological Distress and Food Insecurity among International Students at a Hungarian University: A Post-Pandemic Survey”.
The authors highly appreciate your valuable comments and suggestions, which we believe have significantly improved the quality of the manuscript. Please find the detailed responses below and the corresponding revisions/corrections in the re-submitted files.

Sincerely,

Reviewer 2 Report

Comments and Suggestions for Authors

Reviewer comments

In this study, Hilal et al. aimed to determine the prevalence of psychological distress (depression, anxiety, and stress) among international students at a Hungarian university two years after the COVID-19 outbreak. Additionally, the authors identified some demographic and socioeconomic factors associated with psychological distress and their relationship with food security status.

Kindly, find below my comments for your response.

Abstract

The authors should please make a statement regarding the kind of statistical analyses that were performed.

Line 25-28: The authors should indicate the odds and 95% CI associated with them.

Introduction

The authors should provide a background picture of the streams of income for International students. This is because I think those on Erasmus scholarship will still have regular stream of income and may be impacted less by the pandemic compared to those who are self-funding and had to be doing other jobs.  

Materials and Methods

In the eligibility criteria, did the authors indicate if the participants should be able to speak English? If yes, how about those that came from Francophone countries and were not fluent in English? Did the authors have that as an exclusion criteria? Is it the case that there is a level of English proficiency that every International student in Hungary is supposed to possess prior to the start of their program?

Why were students on Erasmus scholarships excluded? I can understand that they may have access to reliable financial provision during the pandemic. However, why didn’t the authors exclude ALL international students on other forms of scholarships?

Statistical analysis

Line 162: Why was “Shapiro-Wilk test” used when the sample size was more than 50 participants?

Author Response

Dear reviewer,
Thank you for the time and effort you dedicated to reviewing our manuscript “Psychological
Distress and Food Insecurity among International Students at a Hungarian University: A Post-Pandemic Survey”.
The authors highly appreciate your valuable comments and suggestions, which we believe have significantly improved the quality of the manuscript. Please find the detailed responses below and the corresponding revisions/corrections in the re-submitted files.

Sincerely,

Reviewer 3 Report

Comments and Suggestions for Authors

This study quantitatively analyzes the relationship between food security and psychological distress among college students. The authors use food security as the explanatory variable and the psychological distress indices as the dependent variables. However, the following points should be addressed if necessary. First, students who experience psychological distress may also experience a decline in their level of food security, suggesting a simultaneity problem exists. In addition to that, potential bias in the estimation results due to missing variables also exists. If this is the case, it is recommended that food security be treated as an endogenous variable. Second, although multiple psychological stresses are used as explanatory variables, it is important to note that if other variables affect each psychological stress but are not included in the explanatory variables, the error terms in each estimation equation may be correlated. In such cases, the standard errors (i.e., confidence intervals) should be corrected by simultaneous estimation.

Author Response

Dear reviewer,

Thank you for taking the time to review our manuscript “Psychological Distress and Food Insecurity among International Students at a Hungarian University: A Post-Pandemic Survey”.

We truly appreciate all your comments and suggestions. We have carefully considered these comments and tried our best to address every one of them.

Sincerely,

Round 2

Reviewer 3 Report

Comments and Suggestions for Authors

The paper has undergone thorough revision, and no further changes are required.